# The Measure of an Outcome: Comparing Norming and Stacking to Benchmark the Effectiveness of Brain Injury Rehabilitation Services

**DOI:** 10.3390/bs13090705

**Published:** 2023-08-25

**Authors:** Sara D. S. Ramos, Rudi Coetzer

**Affiliations:** 1Brainkind, Wakefield WF5 9TJ, UK; rudi.coetzer@thedtgroup.org; 2Faculty of Health and Medical Sciences, University of Surrey, Guildford GU2 7XH, UK; 3School of Health & Behavioural Sciences, Bangor University, Bangor LL57 2DG, UK; 4School of Psychology, Medicine, Health and Life Science Faculty, Swansea University, Swansea SA2 8PP, UK

**Keywords:** acquired brain injury, clinical evaluation, holistic neurorehabilitation, quality improvement, service development

## Abstract

Practitioners have a clinical, ethical, academic, and economic responsibility to dispassionately consider how effective their services are. Approaches to measure how “good” or “bad” healthcare is include clinical audit, satisfaction surveys, and routine outcome measurement. However, the process of comparing the clinical outcomes of a specific service against the ‘best’ services in the same specialism, also known as benchmarking, remains challenging, and it is unclear how it affects quality improvement. This paper piloted and compared two different approaches to benchmarking to assess clinical outcomes in neurorehabilitation. Norming involved comparing routine measures of clinical outcome with external validators. Stacking involved pooling and comparing internal data across several years. The analyses of routine clinical outcome data from 167 patients revealed significant differences in the patient characteristics of those admitted to the same service provider over time, but no differences in outcomes achieved when comparing with historical data or with external reference data. These findings illustrate the potential advantages and limitations of using stacking and norming to benchmark clinical outcomes, and how the results from each approach might be used to evaluate service effectiveness and inform quality improvement within the field of brain injury rehabilitation.

## 1. Introduction

No matter who provides or funds the care and rehabilitation of patients with acquired brain injury, service providers have a clinical, ethical, scientific, and economic responsibility to dispassionately consider how effective their services are [1]. At the frontline of service delivery, clinicians want to provide the best care for each individual patient seen, independently of how different their diagnoses, impairments, and associated disabilities might be. Similarly, the universal message of healthcare ethical codes is to “do no harm”. Which conversely advocates doing “good”, wherever possible and achievable. Scientists, on the other hand, search for “the truth”, and this is also the case as regards applied clinical research looking at brain injury rehabilitation. Furthermore, everything has a price. If it is the state providing the care, the taxpayer pays; in the case of private healthcare, the patient, their family, or an insurance company must foot the bill. It is, thus, reasonable to expect and search for robust investigative mechanisms to consider the comparative effectiveness of neurorehabilitation [2]. 

It is not always straightforward to measure how good (or bad) the healthcare provided by an organisation is. Some approaches include clinical audit, routine outcome measurement, and patient satisfaction surveys. Clinical audit requires comparison to an agreed standard, which is often difficult to establish in the field of brain injury rehabilitation due to methodological limitations [3], whereas results on routine outcome measures and satisfaction surveys alone may have limited meaningfulness without a context. Benchmarking provides another dimension to measuring, or evidencing, clinical effectiveness in healthcare. Benchmarking can, very simplistically, be defined as the process of comparing a specific service’s clinical outcomes against other services in the same specialism. Wilmington and colleagues [4], in their recent paper, provide an excellent review and analysis of the current state of benchmarking in healthcare. Thus far, benchmarking has had limited uptake by providers of brain injury rehabilitation services. The reasons for this are complex. There is very substantial heterogeneity in the patients seen by rehabilitation services, as regards for example neuropathology, severity, age, time since injury, and comorbidities [5,6,7]. It is extremely difficult to find a valid and reliable comparison group, given the heterogeneity. Not all services routinely collect outcome data. Not all services collect the same outcome data. Different services provide different types of neurorehabilitation, at differing intensities, with variation in clinical staffing levels in teams, and for different lengths of time. Perhaps unsurprisingly, a search of the scientific literature revealed that benchmarking has not proved to be a commonly used method to assessing effectiveness in neurorehabilitation. More specifically, a traditional approach to benchmarking by using third-party outcome data as a comparator has not been used. Instead, the only published research in a UK independent neurorehabilitation provider used their own historical pooled data to benchmark clinical effectiveness [8]. 

Besides the lack of studies explicitly using benchmarking as the main approach to assessing effectiveness of service delivery, there is also the question of which measures would be most suitable to use. In this regard, Alderman and colleagues [9] analysed pooled data from different UK brain injury rehabilitation services using four widely known outcome measures (UK Functional Assessment Measure—FIM+FAM, Mayo-Portland Adaptability Inventory—MPAI-4, Saint Andrews and Swansea Neurobehavioural Outcome Scale—SASNOS, Supervision Rating Scale—SRS). Their findings demonstrated that these four measures are sensitive to change and capable of capturing individual progress through a reduction in impairment and increase in autonomy. 

Benchmarking considering outcome data from the same service or service provider over time periods is still in the early stages of widespread adoption in healthcare [4] but could be a useful tool to improve quality of care. There are several benefits to such a “looking inward”, stacking approach to benchmarking, a technique which involves pooling of existing outcome data within a service across several years and using it as a retrospective comparison group. These include a more closely matched reference sample, with similar patient characteristics, reduced data burden, and avoiding the need for corrections required for risk adjustments [10]. Nevertheless, there are also limitations, including the possibility of not considering external solutions to internal problems and the potential failure to provide a broader context within which to see local trends [10]. 

The alternative to stacking is norming, which consists of reviewing the most up-to-date research literature produced in the field and using that as a reference group. There are several advantages to stacking over norming: it provides an opportunity for time trend analyses, it should provide a (relatively) large sample, against which detection of any cross-sectional change should be possible, and it arguably provides the closest clinical reference group (in contrast to, for example, same sector, but different type of brain injury services). However, the use of stacked data also has limitations, such as high data homogeneity, which may obscure actual change [11]. Put differently and paraphrasing the famous physicist [12], we cannot expect different results from doing the same thing over and over. Furthermore, there is reduced face validity in using within-service data to potentially generalize results externally or widely across the sector. The benefits of norming include potentially large sample sizes, broader comparison across services and countries, and greater potential for generalizability. The limitations include dissimilar clinical populations, or subpopulations, within each comparison group as a whole and limited availability of published results for the intended measures, stratified by relevant clinical characteristics (e.g., diagnosis, injury severity, time since injury). 

In this study, we compared stacking with norming to provide a practical illustration of how the use of these two methods can add an external check and balance to the internal approach of stacking in the analysis of the clinical outcome data from services operated by Brainkind (previously known as BIRT, The Disabilities Trust), a UK-based not-for-profit provider of post-acute brain injury rehabilitation. The aim is to compare how these approaches might reduce some of the technical challenges to benchmarking seen in brain injury rehabilitation, and to identify whether the findings arising from the different techniques increase transparency and inform service improvement and development.

## 2. Materials and Methods

### 2.1. Ethical Considerations

The collection of clinical outcome measures on admission and discharge from rehabilitation within the services provided by Brainkind is part of the standard evaluation policies and procedures, which have been reviewed and approved by Brainkind’s clinical and governance teams. The present work did not raise any of the ethical issues considered within the Health Quality Improvement Partnership Guidance [13] and, in line with the UK Policy Framework for Health and Social Care Research [14], meets the criteria for quality improvement activities exempt from research ethics review. Data processing has been conducted according to the principles expressed in the UK General Data Protection Regulation/Data Protection Act (2018) [15].

### 2.2. Study Design

This study uses a mixed design, combining paired comparisons of routine clinical outcome within individuals over time (between admission and discharge from rehabilitation), cohort comparisons within a single rehabilitation provider over epochs, and cross-sectional comparisons of the present data with published results from other similar patient cohorts.

### 2.3. Participants

Routine clinical outcome data of 167 patients discharged from post-acute brain injury rehabilitation between 1 June 2021 and 31 May 2022 were included in this study. Data were only excluded if the rehabilitation programme had not been completed as planned (early discharge, 19%). Pairwise deletion was used to deal with missing data. The rehabilitation services evaluated in this study were provided by Brainkind’s (BIRT, Brain Injury Rehabilitation Trust) network, which follows a neurobehavioural therapy approach to rehabilitation described by Coetzer and Ramos [16].

### 2.4. Setting

Brainkind’s network of services provides rehabilitation and support to people with an acquired brain injury. Rehabilitation services are indicated for people who are medically stable and within two years post-injury, but who require further assessment and rehabilitation before they are able to return home or live more independently after discharge from hospital. Rehabilitation programmes are tailored to individual needs, after a comprehensive assessment by a transdisciplinary team comprising clinical psychologists, neuropsychologists, occupational therapists, physiotherapists, speech and language therapists, a team of rehabilitation support workers, and neuropsychiatrists and nurses (rehabilitation hospitals). The team works through a neuropsychology-informed approach to deliver individual programmes which vary in content and intensity of sessions on the basis of each individual’s presentation and goals. For example, people who have good self-awareness into their condition and do not experience barriers to rehabilitation, such as severe fatigue or behaviours that challenge, are offered 45 min of the relevant therapies at least five days a week [17]. Programmes also comprise a range of group sessions and leisure and vocational activities, including “Understanding Brain Injury” group, “Memory” group, music therapy, and arts and crafts, among others. For patients who are not able to engage with therapy of this level of intensity, rehabilitation focuses on stabilisation, increasing adjustment and reducing distress. In those situations, input from a neuropsychiatrist may be required, as well as behavioural and psychological interventions delivered by clinical staff (e.g., orientation, mindfulness), or environmental adaptations primarily delivered by rehabilitation support workers (e.g., prompting and support to develop the use of compensatory strategies, consistent responses to specific behaviours with the aim of promoting adjustment and change). The intensity of rehabilitation delivered directly from clinical practitioners may be lower for those who have sustained a brain injury more than two years prior to admission, and the nature of the therapies offered largely depends on changing needs or the prevention or management of crises. 

### 2.5. Measures

We compared outcomes at admission and discharge from rehabilitation, on measures of supervision (Supervision Rating Scale, SRS [18]), impairment, adjustment, and participation (Mayo-Portland Adaptability Inventory-4, MPAI-4, [19]). Although length of stay has been described as an indicator of service efficiency [20], it is shown here alongside rehabilitation outcomes to reflect its complexity and potential to be influenced by a variety of factors, not all of which are within the control of practitioners or of the services within which they operate. The selected measures (SRS and MPAI-4) reflect the primary goals of the rehabilitation approach adopted in these services, are of interest to health and social care systems (e.g., reduction in supervision) and have been validated, recommended, and used to assess outcome and recovery in similar clinical populations (e.g., [21,22]). Normative data were extracted from selected published research on outcomes from post-acute inpatient rehabilitation. We searched publications in the last five years reporting data on at least one of the two outcome measures included in this study at two time points in post-acute rehabilitation. Older studies, or studies where the target measures were reported as means at different time points, or mean change, in charts, rather than as exact scores on tables (e.g., [23,24]) were excluded. As the aim was to illustrate the use of two benchmarking techniques, rather than to conduct a meta-analysis, the number of articles selected for comparison was limited to three. Limiting the number facilitates interpretation of the similarities and differences across the samples. The selected studies were King and colleagues [25], Alderman and colleagues [9], and Jackson and colleagues [26]. 

### 2.6. Data Analysis

Data from the present cohort (Current) were compared with the data gathered within the same services in the previous year (1 June 2020 and 31 May 2021, *N* = 214—Last), and with historical results (1 June 2012 and 31 May 2017, *N* = 770—Historical). Additional comparisons were made (1) between the data from the Current cohort and published reference data and (2) for all three cohorts across three clinical streams [16], which are clinically observed categories used to describe the stage of brain injury recovery and rehabilitation needs of each individual. The three streams are “restoration”, “compensation” and “support”. Restoration is for people who have significant needs in specific areas, for example self-care, communication, or mobility, because of a recent brain injury, and who are likely to benefit from approaches focused on restoration of function. People who benefit from this stream typically have good awareness of their injury, and of how it has affected them, and do not show behaviours of concern [27] or other barriers which would prevent them from taking part in rehabilitation (e.g., refusal of care and treatment). Compensation is for people whose needs may present as barriers to taking part in rehabilitation and who are likely to benefit from approaches primarily focused on developing compensatory strategies to increase function. People in this group may need initial support, prompting, and feedback to become more aware of, and adjust to, the difficulties they face after brain injury. In some cases, patients may have both cognitive and emotional difficulties, as well as physical health needs. The support stream is for people who benefit from ongoing clinical input and support to maintain function and prevent relapse or deterioration. Improvements as a result of functional skills training are achievable but may take longer to come to fruition than the gains seen in those in the restoration or compensation streams [16]. The aims of this comparison were to provide further information on the characteristics of the sample, and to explore the level of change observed within each clinical stream. Parametric statistics were used for normally distributed, interval data, and nonparametric statistics for non-normally distributed ordinal or categorical data, and between-subjects or within-subject techniques were, respectively, used to compare results across cohorts or within the same individuals across time. Given the large number of comparisons, the significance level was set at *p* < 0.01.

## 3. Results

### 3.1. Patient Characteristics

An overview of the patient characteristics across the different stacking groups is shown on Table 1. The current cohort was comparable with the cohort from the previous year and historically in most areas, except for a difference in the male to female ratios (*X*^2^ = 10.26, *df* = 2, *p* = 0.006), which showed a higher proportion of women admitted in the current cohort (*X*^2^ = 8.13, *df* = 1, *p* = 0.004); older age in the current cohort than historically (*t* (1, 932) = 6.11, *p* < 0.001); a higher proportion of people admitted with stroke in the current cohort than historically (*X*^2^ = 9.90, *df* = 2, *p* = 0.007); and a difference in the prevalence of certain comorbidities (drug *X*^2^ = 9.19, *df* = 2, *p* = 0.01, alcohol misuse, *X*^2^ = 22.44, *df* = 2, *p* < 0.001 and multiple trauma, *X*^2^ = 16.16, *df* = 2, *p* = 0.001), which were less prevalent in the current cohort. There was also a difference in time since injury, indicating that people have been admitted on average two months sooner currently than historically (*W* = 41,364, *p* < 0.001). 

### 3.2. Rehabilitation Outcomes

#### 3.2.1. Stacking by Time

Table 2 compares outcomes internally by stacking them by epoch. As length of stay in weeks was skewed, with most people (75%) being discharged within 41 weeks, medians and nonparametric statistics were used for this variable. This showed that people admitted for rehabilitation in the current cohort had shorter stays (*W* = 36,114, *p* = 0.001). Overall, the results show positive change, with significant reductions in the levels of supervision between admission and discharge, and reduction in global disability, impairment, and participation across all epochs (all *p* < 0.01). However, in the current cohort, the difference between admission and discharge for Adjustment was marginal (*p* = 0.06). In addition, a comparison of the proportion of individuals showing clinically significant change on the MPAI-4 indicated that these improvements had marginally decreased in the current cohort for global disability, and in the last two years for participation. 

#### 3.2.2. Stacking by Clinical Stream

Table 3 compares outcomes internally by stacking them by clinical stream. Analyses revealed that lengths of stay were longer for those in the Compensation (*W* = 51,755, *p* < 0.001) and Support streams (*W* = 51,755, *p* < 0.001) compared to those discharged from the Restoration stream. But stays in the Compensation stream were also significantly shorter than those in the Support stream (*W* = 20,317, *p* < 0.001). However, there were significant changes between admission and discharge across the three clinical streams for all variables (all *p* < 0.001). It is also apparent that the scores of those in the Support stream both on admission and on discharge reveal more severe disability compared to those in the Restoration and Compensation streams. Therefore, while there were significant improvements across the three groups, the level of functioning for people discharged from the Support stream was lower. 

#### 3.2.3. Norming

Figure 1 compares the mean change on the SRS and MPAI-4 subscale and total scores observed within our services in the current year, with the mean change reported on King et al. [25], Alderman et al. [9], and Jackson et al. [26]. Statistical comparisons, using MedCalc [28], of the observed mean change between our services and the average change across the three published studies did not reveal significant differences. As individual scores are not available for the published papers, no further statistical comparisons were carried out. However, visual inspection of the data displayed in the figures indicates that effectiveness is comparable to the average or highest change seen in the normative data, except for the MPAI-4 Adjustment, where there may be a trend for a degree of change slightly lower than average in our current cohort. 

## 4. Discussion

This study presents a novel approach set out to compare different methods of benchmarking, including using stacking of internal data, and comparisons with published reference data. We found minor differences in the stacking comparisons, which indicated a shift in the characteristics of people being admitted to rehabilitation in the last two years from those admitted historically. The main differences were on time since injury, which has become shorter; primary diagnosis, which now includes a higher proportion of people with stroke, and older age. Over time, there has also been an increase in the proportion of women admitted and a reduction in specific comorbidities. These do not appear to have influenced outcomes, which were, by and large, comparable across epochs. However, other studies have found that characteristics such as age, stroke severity [7], and level of independence prior to admission [6] were significantly associated with functional outcomes on discharge from rehabilitation.

Comparisons with reference data demonstrated that the target services described in this study are currently achieving outcomes which are at least equivalent to those observed in other national and international services in most areas. However, the area within which the target services performed more modestly, was adjustment and, to a lesser extent, participation. The two cohorts which performed better in these areas were described by Alderman et al. [9] and Jackson et al. [26]. In both cases though, time since injury was significantly longer (37 months in Alderman et al. and 16 in Jackson et al.), as was patients’ length of stay in services (23 weeks in Alderman et al. and 84 in Jackson et al.). This may reflect on the one hand, the timeliness to work around adjustment and participation with patients, which may be best achieved in the medium to longer term after an injury has occurred. On the other hand, it is likely that interventions targeting those areas require much longer times in services for maximum effectiveness. 

The patterns of change observed across the three clinical streams reflect what would be expected given the nature of the problems presented and stage of recovery. Following the neurobehavioural therapy model described in Coetzer and Ramos [16], outcomes should be best for Restoration, as these would be maximised by spontaneous recovery and intensive rehabilitation, and intermediate for Compensation, as this stream is offered to those who present with barriers to intensive rehabilitation, such as severe memory and learning impairment, lack of self-awareness, and behaviours of concern [27]. Support is the stream where outcomes would be expected to be most modest, as this stream is indicated for those in the chronic phase of injury, who present with ongoing complex needs that may relapse and affect quality of life in the longer term (e.g., depression and anxiety which, when aggravated, may result in behavioural problems). The results were consistent with these hypotheses. The best results were observed in the two rehabilitation streams (Restoration and Compensation), whilst there was significant but more limited change in the Support stream.

Brainkind has developed and maintained an outcome measures system and has regularly reported outcome data for many years [29,30,31]. While such results are very useful in their own right, adding a comparative element through a standard or “benchmark” is one way of further increasing our understanding of what as professionals we provide to those we care for, and how cost-effective it is. For example, the findings from this study suggest that the outcomes from the rehabilitation services included in these analyses are similar, or even modestly better, to outcomes reported by comparable service providers, but delivered in a shorter period of time. 

Finding a genuinely “equivalent” group to measure outcome data against is fraught with difficulty. The present study illustrated the use of two broad approaches to benchmarking in brain injury rehabilitation settings. The main findings indicate that comparing outcomes internally and externally can provide some indication as to the level of effectiveness in a particular service, but it may also reveal areas that need investigation and development. For example, we found that the characteristics of the patients admitted to our services have changed over time, reflecting evolving national commissioning and funding patterns. At present, the outcomes appear to be comparable over time, with some differences more attributable to clinical characteristics, such as time since injury and barriers to engagement in rehabilitation. Equally, the comparisons across clinical streams illustrate how different needs and presentations have some influence on the outcomes achieved, providing some indication on required dose and optimal timing for intervention. 

Where our results more closely matched those of published studies, the global characteristics of the samples were also more comparable. The stacking approach, however, revealed more differences on characteristics on admission than on outcomes achieved, which suggests that those differences alone are unlikely to significantly influence the outcomes of an approach to rehabilitation that strives to closely match the interventions to individual needs. 

The main limitation of this study is that it did not carry out a systematic review and meta-analysis of the literature to use as a normative comparator. However, that is an investigation in its own right and is beyond the scope of this paper. Instead, we aimed to present a practical illustration of different benchmarking methods for evaluating routine clinical outcome data in brain injury rehabilitation. It is also acknowledged that the outcome measures selected may not cover all relevant areas in sufficient detail, or that the introduction of other existing or emerging measures, such as a forthcoming adaptation of the Adaptive Behaviour and Community Competency Scale [32] for neurological patients, may be warranted over time. However, this illustration on a large set of clinical data collected over an extensive period of time is one of its key strengths, as this enabled us to explore the extent to which group heterogeneity is likely to influence outcomes and the feasibility of making meaningful comparisons over time and across cohorts with differing characteristics. There is some attrition; not all outcome measures were available for all those regularly discharged from services. Nevertheless, the overall sample size is robust, and the results reflect outcomes achieved in regular service delivery. Finally, the results presented end at the point of discharge from rehabilitation, where outcomes are likely to be at their peak. We cannot but highlight the importance of conducting extended follow-ups to truly demonstrate the effectiveness of rehabilitation, and evaluate its impact on behaviour generalisation, maintenance of functional ability and its possible value to potentiate further learning and wellbeing after rehabilitation has ended. However, while this is practiced within our services to a degree (it is not currently feasible to conduct follow-ups beyond six months post-discharge) [30], the lack of studies with significant follow-up periods remains a gap in the literature. The reasons for this are partly to do with the practical and financial impact of gathering these data, and partly due to the unavoidable high levels of attrition, which are a major limitation in terms of the potential for generalizability of the findings, and therefore potential for publication. 

## 5. Conclusions

The present results suggest that there is no “best” method for benchmarking services, and that the answers we get, and areas for further exploration we uncover, may differ depending on the method selected. Nevertheless, using two different approaches within a benchmarking initiative in a service, is likely to provide a robust system of evaluating results from multiple angles and enable the identification of areas where services excel or can improve.

## Figures and Tables

**Figure 1 behavsci-13-00705-f001:**
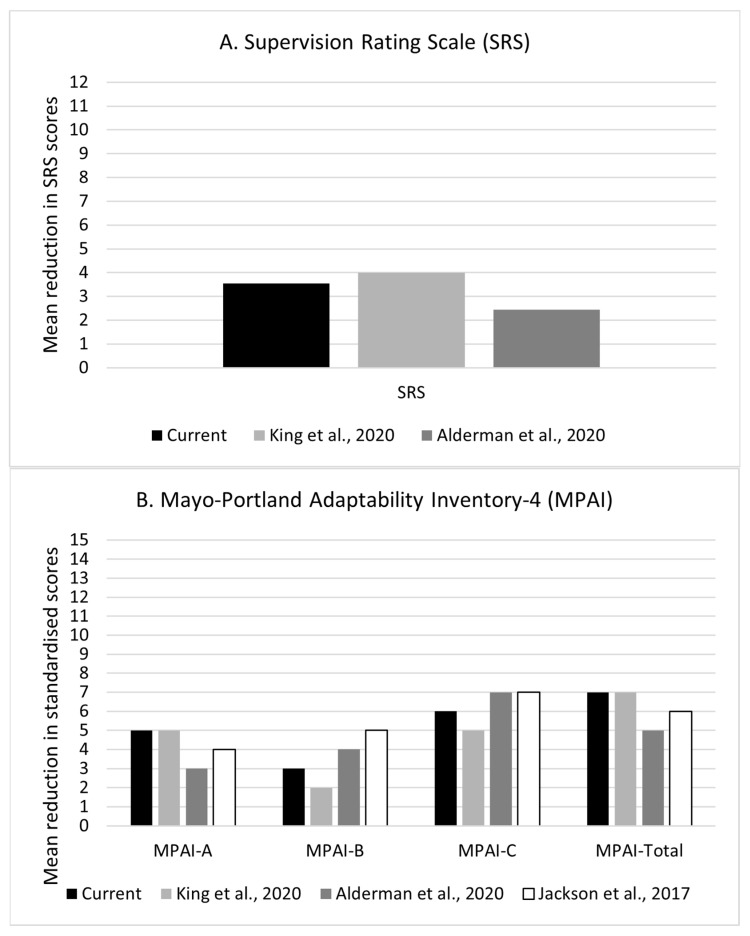
Comparing mean change between two time points with published normative data (**A**) for the SRS (King et al. [25], Alderman et al. [9]) and (**B**) for the MPAI-4 (King et al. [25], Alderman et al. [9], and Jackson et al. [26]).

**Table 1 behavsci-13-00705-t001:** Characteristics of the people admitted across the different comparator groups.

	Current	Last	Historical
Characteristic	(*N* = 166)	(*N* = 214)	(*N* = 770)
**Age** (*M*, *SD*)	57 (18)	57 (16)	48 (15) ‡
**Sex**			
Male	102 (61%)	139 (65%)	556 (72%)
Female	65 (39%)	75(35%)	214 (28%) ‡
**Diagnosis**			
TBI	58 (35%)	72 (34%)	318 (42%)
Stroke	76 (46%)	102 (48%)	251 (33%) ‡
Other	32 (19%)	38 (18%)	192 (25%)
**Months since injury**			
*m* (*IQR*)	2 (1–5)	1 (1–5)	4 (2–10) ‡
**Comorbidities**			
Schizophrenia	4 (3%)	6 (4%)	10 (3%)
Drug misuse	8 (7%)	13 (9%)	50 (16%) ‡
Alcohol misuse	19 (16%)	37 (25%)	124 (38%) ‡
Multiple trauma	7 (8%)	13 (11%)	73 (24%) ‡
Other medical conditions	48 (59%)	67 (64%)	137 (50%)

**Note**. All comparisons are made with reference to Current. ‡ = significantly different (*p* ≤ 0.01). As the characteristics of normative samples differ across measures and are in the public domain as original published research, these will be described and considered in the discussion.

**Table 2 behavsci-13-00705-t002:** Comparing average scores on admission and discharge through stacking by epoch.

	Current		Last		Historical	
Characteristic	(*N* = 137)		(*N* = 167)		(*N* = 647)	
**Weeks in service**						
*m* (*IQR*)	16 (11–34)		16 (10–39)		24 (13–49)	
	A	D	A	D	A	D
**Supervision Rating Scale**						
*N*	94	94	135	135	448	448
*m* (*IQR*)	9 (8–11)	6 (2–8)	9 (8–11)	5 (1–8)	9 (8–11)	6 (2–8)
**MPAI-4 Ability**						
*N*	82	82	141	141	383	383
*M* (*SD*)	53 (11)	48 (11)	52 (8)	46 (11)	52 (9)	47 (9)
% above MCI threshold	-	52%	-	61%	-	59%
**MPAI-4 Adjustment**						
*N*	83	83	147	147	420	420
*M* (*SD*)	54 (10)	51 (10)	52(9)	48 (12)	53 (8)	48 (9)
% above MCI threshold	-	47%	-	54%	-	59%
**MPAI-4 Participation**						
*N*	83	83	144	144	433	433
*M* (*SD*)	59(13)	52 (13)	57(11)	51 (11)	57 (10)	49 (10)
% above MCI threshold	-	55%	-	56%	-	68% †
**MPAI-4 Total**						
*N*	79	79	138	138	348	348
*M* (*SD*)	61 (14)	55 (15)	58 (10)	51 (13)	58 (10)	51 (11)
% above MCI threshold	-	53% †	-	66%	-	68%

**Note**. MCI = Minimal Clinically Important Difference. † = marginally different (0.01 < *p* ≤ 0.06).

**Table 3 behavsci-13-00705-t003:** Comparing average scores on admission and discharge through stacking by clinical stream.

	Restoration		Compensation		Support	
Characteristic	(*N* = 434)		(*N* = 356)		(*N* = 145)	
**Weeks in service**						
*m* (*IQR*)	12 (8–21)		21 (12–40) ‡		31 (16–57) ‡	
	A	D	A	D	A	D
**Supervision Rating Scale**						
*N*	359	359	284	284	110	110
*m* (*IQR*)	8 (6–11)	5 (1–7)	9 (7–11)	5 (2–8)	10 (8–11)	8 (7–10)
**MPAI-4 Ability**						
*N*	350	350	275	275	104	104
*Mean* (*SD*)	51 (9)	45 (10)	50 (8)	44 (10)	52 (10)	53 (8)
% above MCI threshold	-	64%	-	59%	-	55%
**MPAI-4 Adjustment**						
*N*	359	359	281	281	106	106
*M* (*SD*)	51 (9)	45 (11)	54 (8)	49 (10)	57 (9)	53 (10)
% above MCI threshold	-	56%	-	55%	-	53%
**MPAI-4 Participation**						
*N*	360	360	279	279	111	111
*M* (*SD*)	54(10)	47 (11)	56(10)	50 (10)	65 (10)	58 (11)
% above MCI threshold	-	63%	-	61%	-	61%
**MPAI-4 Total**						
*N*	334	334	264	264	96	96
*M* (*SD*)	56 (11)	47 (12)	57 (10)	50 (12)	67 (12)	59 (13)
% above MCI threshold	-	69% †	-	64%	-	59%

**Note**. MCI = Minimal Clinically Important Difference. † = marginally different (0.01 < *p* ≤ 0.06) ‡ = significantly different *p* ≤ 0.01.

## Data Availability

The authors do not have permission to share the data publicly due to restrictions under UK data protection regulations and Brainkind’s Privacy Policy.

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
