# Peer review of "The Measure of an Outcome: Comparing Norming and Stacking to Benchmark the Effectiveness of Brain Injury Rehabilitation Services"

_behavsci, 2023, doi:10.3390/bs13090705_

Round 1

Reviewer 1 Report

I would define in more detail the type of Rehabilitation received (physioterapy, occupational therapy, speech therapy), and the intensity of each treatment.

I would recommend more recent bibliography and more focused on functional measures in Rehabilitation.

Examples of updated references:

Vázquez-Guimaraens, M.; Caamaño-Ponte, J.L.; Seoane-Pillado, T.; Cudeiro, J. Factors Related to Greater Functional Recovery after Suffering a Stroke. Brain Sci. 2021, 11, 802

Stinear, C.M.; Byblow, W.; Ackerley, S.J.; Barber, P.A.; Smith, M.C. Predicting Recovery Potential for Individual Stroke Patients Increases Rehabilitation Efficiency. Stroke 2017, 48, 1011–1019

Salvadori, E.; Papi, G.; Insalat, G.; Rinnoci, V.; Donnini, I.; Martini, M.; Falsini, C.; Hakiki, B.; Romoli, A.; Barboto, C.; et al. Comparison between Ischemic and Hemorrhagic Strokes in Functional Outcome Discharge from an Intensive Rehabilitation Hospital. Diagnostics 2020, 11, 38. 

Anaya, M.A.; Branscheidt, M. Neurorehabilitation After Stroke. Stroke 2019, 50, e180–e182.

Author Response

Point 1: “I would define in more detail the type of Rehabilitation received (physioterapy, occupational therapy, speech therapy), and the intensity of each treatment.”

Response 1: The Disabilities Trust provides a wide range of services covering the full rehabilitation pathway following brain injury. As interventions are tailored to the stage of recovery and are adapted to each individual according to their needs, goals and ability to engage and participate in rehabilitation, the nature and intensity of care and treatment will vary. We have added a new section (2.4 Setting, ln. 135-159) to provide more detail about how the intensity and nature of “rehabilitation” differ, depending on individual needs and across the different stages of recovery.

Point 2: “I would recommend more recent bibliography and more focused on functional measures in Rehabilitation.

Examples of updated references:

Vázquez-Guimaraens, M.; Caamaño-Ponte, J.L.; Seoane-Pillado, T.; Cudeiro, J. Factors Related to Greater Functional Recovery after Suffering a Stroke. Brain Sci. 2021, 11, 802

 Stinear, C.M.; Byblow, W.; Ackerley, S.J.; Barber, P.A.; Smith, M.C. Predicting Recovery Potential for Individual Stroke Patients Increases Rehabilitation Efficiency. Stroke 2017, 48, 1011–1019

Salvadori, E.; Papi, G.; Insalat, G.; Rinnoci, V.; Donnini, I.; Martini, M.; Falsini, C.; Hakiki, B.; Romoli, A.; Barboto, C.; et al. Comparison between Ischemic and Hemorrhagic Strokes in Functional Outcome Discharge from an Intensive Rehabilitation Hospital. Diagnostics 2020, 11, 38.

Anaya, M.A.; Branscheidt, M. Neurorehabilitation After Stroke. Stroke 2019, 50, e180–e182.”

Response 2: We have added the suggested references that best exemplified the potential and complexity of  rehabilitation outcomes research (ln. 57, 282-285). As over half of the sample in our study had an acquired brain injury from causes other than stroke, and there are significant differences in patient characteristics compared with the populations described in a lot of the stroke rehabilitation literature (e. g. neurobehavioural rehabilitation patients tend to be younger), we have kept the references originally included, which focus on populations that more closely resemble that seen in our services, and services which follow a similar rehabilitation approach.

Reviewer 2 Report

Useful positioning of the paper in the introduction, and the potential schism between “truth seekers” (scientists) and “do good/do no harm” clinicians. I think that having an experienced, dedicated and specialised clinician and an experienced, dedicated and specialised researcher work together, is a great starting point in terms of authors. 

There is potential to mention the Adaptive Behaviour and Community Competency Scale as an outcome measure. (Giles GM. Assessing adaptive behaviour in the post-acute setting following traumatic brain injury: initial reliability and validity of the Adaptive Behavior and Community Competency Scale (ABCCS). Brain Injury 2007;21:521–529) Giles et al are in the process of updating this to a neuro-functional outcome scale which is not yet published. 

The issue with measurements taken in the rehabilitation setting is that, even if quite good and with good inter-rater reliability, they assess functioning in the heavily scaffolded setting of the rehabilitation setting, they do not assess intrinsic skills, or the degree to which compensatory strategies are internalised and generalised across time and setting. They are environmentally mediated, and that environment includes the expectations, routines and embedded cueing that occurs in externally governed settings. One could reasonably argue that effectiveness of rehabilitation is measured by how someone is functioning 1, 2, 10 years post discharge. However that too is mediated by environmental factors and longer term support (or not) over which a rehabilitation unit has no control.

Line 138: length of stay may be best described as an indicator of service efficiency 

This concept troubles me. Because LOS may actually be indicative of a huge number of factors that are intrinsic to the individual, their premorbid state, their impairments, the interrelationship between the impairments, and also to the service, and the political and economic circumstances that the service exists within. I think that this sentence could be clearer.

The clinical streams could be better described in lay terms, what do they each mean? 

There is no mention of factors extrinsic to the rehabilitation unit’s control, that of funders’ eligibility criteria, and access to funds for individuals to facilitate inpatient rehabilitation, this is very important as this changes over time. Amount of supervision is a function of what will be paid for, not simply a function of what is required. These difficulties also have an impact upon LOS, as does factors such as the ability to access highly skilled community services to facilitate planned and graded discharge and maintain the gains from the rehab setting. As rehabilitation is surely also related to staff skills and experience, this has an impact upon service provision and quality of service. Recent difficulties with recruitment are, in my experience, having a serious and detrimental impact upon quality of inpatient post-acute neurorehabilitation, and is having a devastating impact upon post-discharge service construction and management: how do we maintain gains across settings when I think we have evidence that quality of staffing is variable, and likely deteriorating? This is particularly the case with the staff that spend the longest with the service users, the rehabilitation assistants/support workers.

As the paper stands currently, I read it as if nothing external has changed, which could impact upon measurement of effectiveness of service, when in fact these are key to how ongoing slow stream rehabilitation is managed, particularly at the time of discharge to the community. 

Having known the BIRT services (and worked alongside them) for more than 2 decades, they themselves are not uniform in terms of provision, and so I am unclear how comparing inside of the service, but across units, is straightforward. Some are far more behavioural than others. When comparing outside of the services, as a person who commissions post-acute inpatient services, a number of factors affect the choice made and factors such as who is the lead clinician and what is the patient profile presently like, affect the decisions, which in turn affects the running of the service. Whether we like it or not, average services can improve, excellent services can deteriorate, and, if you have been doing it as long as I have (!) cycles of average to excellent, to poor, to average, to excellent are observable.  

Overall, I think that the paper is correct to look for ways of benchmarking against other services, and against the same service over time, but I think that the heterogeneity problems need to be more foregrounded, as do the extrinsic cultural/political/economic and social changes, as does the fact that what is being measured is the person, in the unit, not the person, in the community post discharge. 

Author Response

Point 1: “Useful positioning of the paper in the introduction, and the potential schism between “truth seekers” (scientists) and “do good/do no harm” clinicians. I think that having an experienced, dedicated and specialised clinician and an experienced, dedicated and specialised researcher work together, is a great starting point in terms of authors.”

Response 1: We would like to thank the reviewer for their positive feedback.

Point 2: “There is potential to mention the Adaptive Behaviour and Community Competency Scale as an outcome measure. (Giles GM. Assessing adaptive behaviour in the post-acute setting following traumatic brain injury: initial reliability and validity of the Adaptive Behavior and Community Competency Scale (ABCCS). Brain Injury 2007;21:521–529) Giles et al are in the process of updating this to a neuro-functional outcome scale which is not yet published.”

Response 2: We have added a sentence in the discussion (ln. 322-326) to acknowledge the importance of reflecting on the performance of any outcome measures used over time, and the potential need for adding to, or replacing, those with more appropriate measures.

Point 3: “The issue with measurements taken in the rehabilitation setting is that, even if quite good and with good inter-rater reliability, they assess functioning in the heavily scaffolded setting of the rehabilitation setting, they do not assess intrinsic skills, or the degree to which compensatory strategies are internalised and generalised across time and setting. They are environmentally mediated, and that environment includes the expectations, routines and embedded cueing that occurs in externally governed settings. One could reasonably argue that effectiveness of rehabilitation is measured by how someone is functioning 1, 2, 10 years post discharge. However that too is mediated by environmental factors and longer term support (or not) over which a rehabilitation unit has no control.”

Response 3: We have added a sentence in the discussion (ln. 332-342) to acknowledge the lack of long-term follow-up as a limitation of this study, and more generally within the field.

Point 4: “Line 138: length of stay may be best described as an indicator of service efficiency

This concept troubles me. Because LOS may actually be indicative of a huge number of factors that are intrinsic to the individual, their premorbid state, their impairments, the interrelationship between the impairments, and also to the service, and the political and economic circumstances that the service exists within. I think that this sentence could be clearer.”

Response 4: We agree with the reviewer’s concern in terms of taking length of stay as a straightforward measure of efficiency, as it is indeed affected by many factors, not all of which are within the control of practitioners or of the services within which they operate. Although we were citing Lowe et al. (2022), we have edited the sentence to reflect these important considerations.

Point 5: “The clinical streams could be better described in lay terms, what do they each mean?”

Response 5: We have added a brief description of each of the three clinical streams in lines 188-204.

Point 6: “There is no mention of factors extrinsic to the rehabilitation unit’s control, that of funders’ eligibility criteria, and access to funds for individuals to facilitate inpatient rehabilitation, this is very important as this changes over time. Amount of supervision is a function of what will be paid for, not simply a function of what is required. These difficulties also have an impact upon LOS, as does factors such as the ability to access highly skilled community services to facilitate planned and graded discharge and maintain the gains from the rehab setting. As rehabilitation is surely also related to staff skills and experience, this has an impact upon service provision and quality of service. Recent difficulties with recruitment are, in my experience, having a serious and detrimental impact upon quality of inpatient post-acute neurorehabilitation, and is having a devastating impact upon post-discharge service construction and management: how do we maintain gains across settings when I think we have evidence that quality of staffing is variable, and likely deteriorating? This is particularly the case with the staff that spend the longest with the service users, the rehabilitation assistants/support workers.

As the paper stands currently, I read it as if nothing external has changed, which could impact upon measurement of effectiveness of service, when in fact these are key to how ongoing slow stream rehabilitation is managed, particularly at the time of discharge to the community”.

Response 6: Following Reviewer 2’s suggestions, the paper now includes more discussion of the potential impact of a multiplicity of external factors on outcomes, which may be beyond the control of practitioners and service providers (lines 165-167, 304-306, 357-360).

Point 7: “Having known the BIRT services (and worked alongside them) for more than 2 decades, they themselves are not uniform in terms of provision, and so I am unclear how comparing inside of the service, but across units, is straightforward. Some are far more behavioural than others. When comparing outside of the services, as a person who commissions post-acute inpatient services, a number of factors affect the choice made and factors such as who is the lead clinician and what is the patient profile presently like, affect the decisions, which in turn affects the running of the service. Whether we like it or not, average services can improve, excellent services can deteriorate, and, if you have been doing it as long as I have (!) cycles of average to excellent, to poor, to average, to excellent are observable.” 

Response 7: We completely agree with this point from Reviewer 2. The aim of having a standardised and systematic outcome measures system within our services is precisely to ensure that we are able to make comparisons, observe differences, and notice deterioration or improvement in performance as it occurs, as well as enabling us to identify the factors that may influence those. It was the insight obtained from studying our outcome results that led us to better understand differences across services, groups of patients, and to evolve from a system based on “types of services” to a system  based on patient needs (i. e. clinical streams). The comparison of the results across streams illustrates this. Comparisons between individual services is, however, beyond the scope of this paper, because the numbers within some of the services prevent reliable comparisons, but also because we aim to show how a system of outcomes used in standard clinical practice for routine service evaluation, can also inform the field of rehabilitation more widely, and lead to the generalisation of results, and to an increased understanding of the outcomes following brain injury rehabilitation and of the factors that may or may not affect those.

Point 8: “Overall, I think that the paper is correct to look for ways of benchmarking against other services, and against the same service over time, but I think that the heterogeneity problems need to be more foregrounded, as do the extrinsic cultural/political/economic and social changes, as does the fact that what is being measured is the person, in the unit, not the person, in the community post discharge.

Response 8: We have revised the paper to bring the challenges of benchmarking, and limitations of this paper, including the lack of extended follow-up, more to the foreground.

Reviewer 3 Report

This article addresses interesting and inovative topic-benchmark the effectiveness of brain injury rehabilitation service. The topic selected by authors is attractive and not enough present as research subject in rehabilitation.

Abstract is clear, informative and give us clear insight in topic and research results presented in this manuscript.

In the Introduction authors introduced us with different options to measure effectiveness of neurorehabilitation as well as benchmarking by norming and stacking. They also explored potential weakness of both methods. My only remark would be regarding aim of the study. From my point of view, the aim cannot be “to evaluate…” but “to compare…”. With that change it is going to be consistent with presented conclusions.

Material and methods are thoroughly presented, including ethical consideration, precise study design and participants.  Measures are well chosen, sufficient, accurate for pathology and efficient for following neurorehabilitation outcomes.

Results are presented clearly, in text and tables, regarding patients and rehabilitation outcomes, divided two groups stacking by time and clinical stream and norming.

Discussion underlines differences between this two approaches, benefits and limitation of both, as well as study limitations. All statements are well documented by appropriate literature. findings. However, I hope that is just a beginning of further exploration of this topic by authors and that they are going to offer us some more conclusive results in future.

Conclusion is consistent with results, no need to have references in conclusion.

Author Response

This article addresses interesting and inovative topic-benchmark the effectiveness of brain injury rehabilitation service. The topic selected by authors is attractive and not enough present as research subject in rehabilitation.

Abstract is clear, informative and give us clear insight in topic and research results presented in this manuscript.

In the Introduction authors introduced us with different options to measure effectiveness of neurorehabilitation as well as benchmarking by norming and stacking. They also explored potential weakness of both methods. My only remark would be regarding aim of the study.

Point 1: From my point of view, the aim cannot be “to evaluate…” but “to compare…”. With that change it is going to be consistent with presented conclusions.

Material and methods are thoroughly presented, including ethical consideration, precise study design and participants. Measures are well chosen, sufficient, accurate for pathology and efficient for following neurorehabilitation outcomes.

Results are presented clearly, in text and tables, regarding patients and rehabilitation outcomes, divided two groups stacking by time and clinical stream and norming.

Discussion underlines differences between this two approaches, benefits and limitation of both, as well as study limitations. All statements are well documented by appropriate literature. findings. However, I hope that is just a beginning of further exploration of this topic by authors and that they are going to offer us some more conclusive results in future.

Point 2: Conclusion is consistent with results, no need to have references in conclusion.”

As Reviewer 3's points for revision are embeded in a broader evaluation and commentary of the paper, we have combined our response to both below.

Firstly, we would like to thank Reviewer 3 for the positive feedback on the paper and for the valuable suggestion to make it the starting point of further exploration of this topic. We hope that we will be able to contribute with more data informed suggestions in future, and increase our ability to not only describe the outcomes of brain injury rehabilitation, and the factors that affect them, but to also explore how that knowledge might be used to adapt and improve rehabilitation processes and achieve better outcomes.

Response 1: We have revised the text in line with the suggestion to change the aim from “evaluate” to “compare”  (ln. 106).

Response 2: We have also removed the references from the conclusion.